# Predictive Quantization and Symbolic Dynamics

Shlomo Dubnov

Music, Computer Science and Engineering, University of California San Diego, La Jolla, CA 92093, USA; sdubnov@ucsd.edu

**Abstract:** Capturing long-term statistics of signals and time series is important for modeling recurrent phenomena, especially when such recurrences are a-periodic and can be characterized by the approximate repetition of variable length motifs, such as patterns in human gestures and trends in financial time series or musical melodies. Regressive and auto-regressive models that are common in such problems, both analytically derived and neural network-based, often suffer from limited memory or tend to accumulate errors, making them sensitive during training. Moreover, such models often assume stationary signal statistics, which makes it difficult to deal with switching regimes or conditional signal dynamics. In this paper, we describe a method for time series modeling that is based on adaptive symbolization that maximizes the predictive information of the resulting sequence. Using approximate string-matching methods, the initial vectorized sequence is quantized into a discrete representation with a variable quantization threshold. Finding an optimal signal embedding is formulated in terms of a predictive bottleneck problem that takes into account the trade-off between representation and prediction accuracy. Several downstream applications based on discrete representation are described in this paper, which includes an analysis of the symbolic dynamics of recurrence statistics, motif extraction, segmentation, query matching, and the estimation of transfer entropy between parallel signals.

**Keywords:** symbolic dynamics; discrete representation learning; predictive information bottleneck; variable Markov oracle; music information dynamics





## 1. Introduction

In this paper, we describe a method for time series symbolization that allows novel applications for various types of signals, such as audio, gestures, and more. Symbolization involves the transformation of raw time series measurements into a series of discretized symbols that are processed to extract information about the generating process. In this paper, the problem of symbolization is formulated in information theoretical terms as a information dynamics bottleneck that optimizes a trade-off between the representation fidelity and predictive quality of the latent discrete representation. The algorithm developed in this paper performs an embedding of the data in a vector space followed by a step of quantization that maximizes a measure that we call information rate that uses approximate string-matching methods to evaluate the mutual information between the signal in the past and present. The move to symbolic data allows for capturing long-term structures that are hard to model using real-valued autoregressive distribution estimation methods. Analysis of data using symbolic dynamics allows the characterization of the time series by various analysis methods, such as recurrent quantification analysis, motif finding, segmentation, and querying, as well as novel methods for transfer entropy estimation between pairs of sequences.

There are multiple works that describe the symbolization of time series. A general feature of symbolic dynamics analysis is that it assumes the existence of so-called generating partitions that divide the phase space of the time series trajectory into regions, such that each unique trajectory in phase space is associated with a unique sequence of symbols. Generating partitions can be constructed for certain models; however there is no general

approach for constructing generating partitions from observations of an unknown system. Accordingly, in the model-free case, various data aggregations techniques are used to cluster nearby data points to induce such partitions. One such popular method, largely due to its simplicity and intuitive nature, is the so-called "threshold-crossing" technique. It operates by replacing real-valued data with symbolic indices based on the data values falling within a certain numeric interval. In extreme cases, a single threshold is used to create a binary representation with values above and below thresholds assigned 1's and 0's, respectively. An example of multi-threshold symbolization is the symbolic aggregate approximation (SAX) [1], shown in Figure 1.

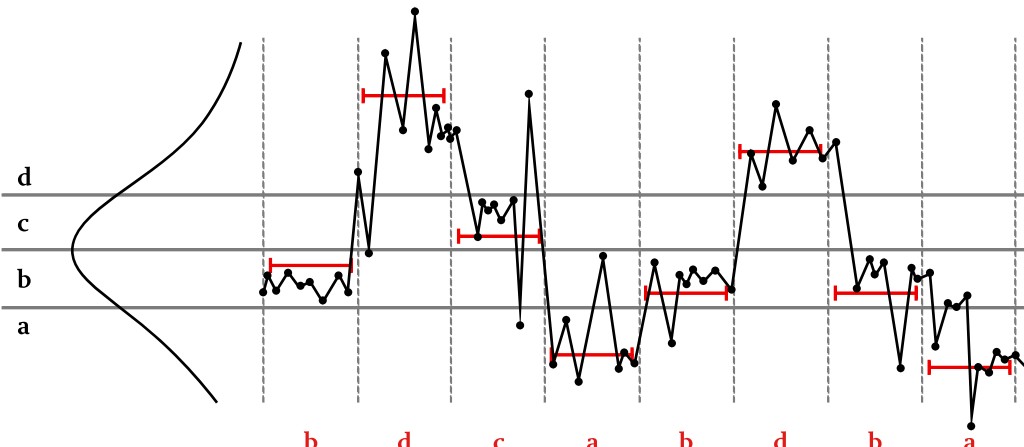

**Figure 1.** Transforming a real-valued continuous or sampled time function into SAX representation with alphabet a, b, c, d. Each time segment is turned into a symbol by taking the mean and looking up the corresponding symbol on y-axis. The complete signal is translated into a word eight letters long, as shown on the bottom of the graph.

In the SAX approach, a scalar quantization is performed along an amplitude axis using an average value over a fixed time interval. First, the number of letters $\alpha$ and maximal sequence length $\omega$ are manually chosen so as to have an overall small number of symbols while not losing too much data detail. The quantization levels are found by fitting a normal distribution to the data amplitude distribution aggregated over time and across multiple data instances. Since the method is meant to be applied for time series databases, for a given $\omega$, some observations may not be $\omega$ letters long.

It is evident from this example that the choice of partition levels is paramount to establishing a correspondence between the continuous-valued data and their symbolic representation. As an extreme example, for a binary partition, setting a threshold outside of the range of the data values will result in a trivial constant symbolic sequence. Errors in coarse-grained partition relative to original system dynamics can lead to a gross misrepresentation of the dynamical system. In particular, one way to demonstrate sensitivity to threshold change is by mapping the system dynamics into a graph structure where each edge carries a label from the symbolic representation, and the vertices are labeled by all possible words in a sequence of a certain length. Changing the partitioning threshold effectively relabels the transitions between the possible words, changing the probabilities of transitioning between the vertices or even forbidding some words from appearing. This effect can be measured by considering the entropy over the word space and comparing it to theoretical entropy in the case when an analytic equation of the system dynamics is known. As shown in Refs. [2,3], the entropy of such language is often lower than the exact system entropy. Moreover, the symbolic sequence entropy dependence on threshold displacement is a non-monotonic function because some words that disappear for some threshold values can reappear at other values.

Rather than operating directly on the data values, some methods of signal or time series analysis perform transformations of the data into more convenient or meaningful vector spaces. It is evident that a suitable choice of such representation is critical for the success of downstream applications, such as classification, data mining, or generative models. When sequence dynamics is relatively simple, transforms, such as DFT, can be used to reveal periodic properties, even if slight deviations from the transform principal patterns occur in the signal. To address the time varying properties of a signal, transformation and change of representation is often done over short segments of data. In special cases, such as in audio signals, custom human-engineered features, such as mel frequency cepstral coefficients (MFCC) or Chroma are often used. When high-dimensional feature representation is efficiently translated into a low-dimensional space, the term "embedding" is used to denote such reduced representation. From a practical perspective, embedding serves as a unique identifier of a short data segment, represented as a vector in some space that maps similar or related data segments to nearby points. The situation is further complicated by longer-term signal dynamics that exhibit a-periodic repetitions with motifs of variable length with partial and approximate repetitions. Combining embeddings with symbolic discretization opens up the study of time series statistics to methods of information and dynamics theory, automata theory, and other formal tools coming from computational modeling. Among the common discretization methods are K-Means and a variety of so-called vector quantization (VQ) methods in general that assign the same index or letter symbol to samples falling within the same multi-dimensional partition. The K-Means algorithm is based on the iterative partitioning of points into regions that are then matched to their centroids and re-partitioned again, with the process repeating until convergence. Self-organizing maps (SOM) are another popular data analysis and visualization tool based on neural networks that bear close similarity to autoencoders (AE) that can learn a non-linear embedding of the data. Clustering can be performed on the SOM nodes or AE latent codes to identify groups of data points with similar metrics.

Recently, generative deep neural models have become the primary tools for representation learning in multiple domains. The compelling idea of machine learning is that various modeling and inference tasks, including embeddings, representation, and encodings of data collected from the world, are captured using vector spaces that map statistical structures into mathematical space. In particular, the powerful idea of generative modeling is that in order to effectively represent the data, the learning system needs to be able to effectively reproduce similar data. Mathematically, this means that the goal of learning is to be able to approximately simulate the statistics of the data from an internal representation that the learning system constructs. Once such statistical representation is constructed, various operations can be performed directly on the model's parameters rather then on the data themselves. For instance, probabilistic clustering is done by an iterative procedure called expectation maximization (EM), which bears close relation to a variational method called evidence lower bound (ELBO), which was adopted using the reparametrization method in variational autoencoders (VAE). The VAE framework represents the data in terms of the multidimensional Gaussian distribution of latent factors, from which the data samples can be recovered. Although a detailed explanation of VAE architecture and its learning procedure by maximizing ELBO is beyond the scope of this paper, we show the general architecture of such a model in Figure 2 .

The important components of the VAE method is that it assumes a probabilistic representation of the latent space, which is learned by an encoder followed by estimating the mean and variance of the encodings as Gaussian distribution. Reconstruction of the data is done by sampling from a normal distribution and "reparametrizing it" by multiplying it with appropriate variance, adding a mean shift to the random sample, and passing it through a learned decoder. The learning process comprises the network parameters so as to achieve a minimization of two criteria, namely the reconstruction loss at the output of the decoder, and a statistical loss written in terms of Kullback–Leibler statistic divergence between the estimated Gaussian distribution of the latent vectors and a prior normal

distribution. This second loss can also be viewed as a regularization penalty, trying to find the simplest random distribution of the latent encoding by trying to make it fit a normal distribution. Recently, VAE representation was further extended into a quantized version named VQ-VAE, becoming a powerful and popular alternative method for so-called "discrete neural representation". In Figure 3, an audio signal is encoded into a series of codes at three different resolutions as part of a larger system that uses transformers to generate high-fidelity and diverse musical materials that are several minutes long [4].

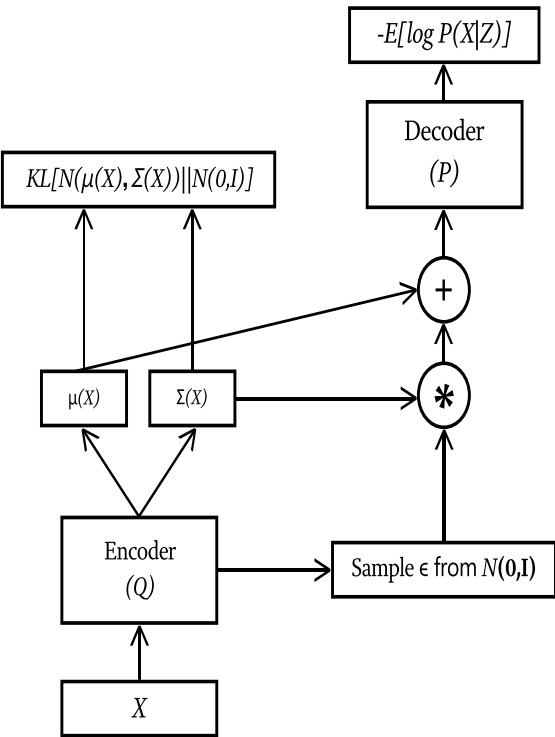

**Figure 2.** VAE architecture showing the encoder–decoder structure with estimation of the mean and variance using the normal distribution reparametrization trick, and the two loss functions components of the ELBO criterion. See text for more details.

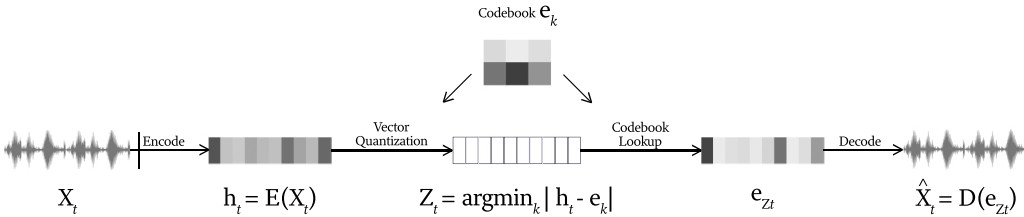

**Figure 3.** VQ-VAE discrete neural architecture that uses codes to represent and reconstruct audio.

The input audio is segmented and encoded into latent vectors $h_t$, which are then quantized to the closest codebook vector $e_k$. The decoder then reconstructs the audio from a sequence of codebook vectors.

In the following, we describe a different method of quantization and symbolization that is based on a lossy encoding of VAE latent codes. Unlike existing methods, where a quantization procedure is used to directly obtain a discrete or symbolized version of the data, the method proposed in this paper focuses on symbolization that maximizes the predictive information of the symbolic sequence. This predictive information, defined in terms of mutual information between the past and the present of the time series, is measured by "information rate". By taking the information dynamics into consideration, we hope to obtain a symbolization that better matches the dynamics of the original data. Limiting

the amount of information in data representation based on predictive criteria is sometimes called a "predictive bottleneck". Moreover, prior to the symbolization step, the complexity of the continuous representation is reduced by applying a bit-rate reduction. This lossy encoding helps remove noise from the continuous latent vectors by allocating fewer bits or even completely removing part of the encoding dimensions that carry little information about the data. During the symbolization step, the amount of predictive information depends on a threshold parameter that is used for distinguishing between the similar versus distinct latent vectors. Analogous to the threshold-crossing method described above, the information rate dependence on a threshold parameter is a non-monotonic function, with a zero information rate occurring at very low and very high thresholds and with several local maxima possible at intermediate threshold values. We use a threshold value that gives the global maximum of information rate as a symbolization choice. An additional use of lossy encoding is in exploring transfer entropy between simultaneous time series. The intuition is that causal relations between separate data streams that originate from related sources might be better revealed by reducing the fidelity of their data representation. To summarize the concepts and methods presented in the paper, in Section 2, we define the general framework of information dynamics; in Section 3, we describe the predictive quantization and time series symbolization using information rate; in Section 4, we discuss representation learning and the use of bit allocation for rate reduction; in Section 5, we develop a symmetric transfer entropy estimator using information rate and mutual information measures; and in Section 6, we describe applications for motif finding and a symbolic version of recurrent quantification analysis.

## 2. Information Dynamics

Information dynamics considers the information passing between the data past $X$ and the present observation $Y$, formally defined in terms of mutual information $I(X, Y)$. In this paper, we extend the notion of information dynamics to aspects of data representation by providing a theoretical model and an analysis framework for exploring the relations between four factors in a time series: the data past $X$ and a present observation $Y$ versus their representation in terms of latent variables $Z$ and $T$, respectively.

This relationship between the four factors $X$, $Y$, $Z$, and $T$ is illustrated in Figure 4, schematically shown for an example of musical observations, with latent states schematically drawn near a brain-like icon. It should be noted that this representation is schematic only—there is no assurance that the latent representation follows similar dynamics to the data or that later on its symbolized version induces correct generating partitions. Nevertheless, we assume that finding a representation that maximizes the information dynamics of the resulting sequence is intuitively a good approach to discover meaningful structures present in the original data. We further assume that the following Markov relations exist between the variables: $Z$ is assumed to be a hidden cause of $X$, which in turn causes $Y$, which is further encoded into $T$. In other words, the triplet $X, Y, Z$ obeys the following generative Markov relation $p(X, Y, Z) = p(Y|X)p(X|Z)p(Z)$, and similarly $p(X, Y, Z, T) = P(T|Y)p(Y|X)p(X|Z)p(Z)$ includes the probability for the next encoding.

The significance of the Markov relation is that it specifies the assumptions about how information in a time series evolves according to the manner in which probabilities of the different variables depend on each other. Based on the Markov relation between $Z - X - Y$, our model comprises two competing factors: quality of data representation in terms of latent variables versus the prediction ability of the latent variables in time. Our model is defined as a minimization problem of the discrepancy between data prediction based on complete information about the past observation $X$ versus prediction based on the latent encoding of the past $Z$. This discrepancy is averaged over all possible encoding pairs $X, Z$, which is stated as follows:

**Lemma 1.** *For Markov relation* $Z - X - Y$, *the following relation holds*

$$\langle D_{KL}(p(Y|X)||p(Y|Z))\rangle_{p(X,Z)} = I(X, Y|Z) = I(X, Y) - I(Z, Y) \qquad (1)$$

*where, $D_{KL}(\cdot,\cdot)$ is the Kullback–Liebler (KL) divergence and $I(\cdot,\cdot)$ is mutual information.*

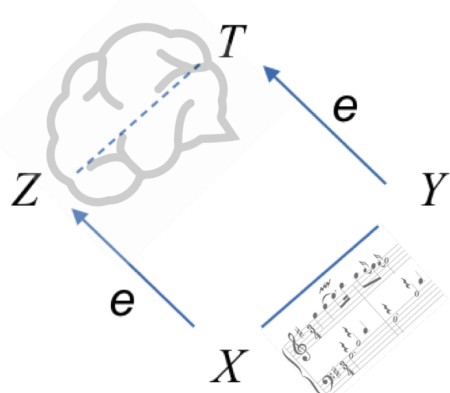

**Figure 4.** Schematic graph of statistical dependencies between the different model variables. The letter "e" represents an embedding created by encoding. Later on we will use VAE as our encoding method. The complexity of the embedding will be controlled by bit-rate allocation. The resulting dynamics of the latent representation are not assured to be in exact correspondence to the dynamics of the data.

**Proof.** From the definitions of $D_{KL}(\cdot,\cdot)$ as the Kullback–Liebler distance between different distributions and $I(\cdot,\cdot)$ as the mutual information between their random variables, we write

$$D_{KL}(P,Q) = \int p(x) \log \frac{p(X)}{q(X)} dX \tag{2}$$

and

$$I(X,Y) = H(X) - H(X|Y) = H(Y) - H(Y|X) = H(X) + H(Y) - H(X,Y) \tag{3}$$

with signal entropy given by

$$H(X) = - \int p(X) \log p(X) dX \tag{4}$$

Using the following relation between KL distance and mutual information

$$I(X,Y) = D_{KL}(p(X,Y), p(X)p(Y)) = D_{KL}(p(Y|X)p(X), p(X)p(Y)), \tag{5}$$

and taking into account the Markov relations $Z - X - Y$, we have the following conditional independence between the variables $p(Y,X,Z) = p(Y|X,Z)p(X,Z) = p(Y|X)p(X,Z)$. Averaging over $P(X,Z)$, we obtain

$$\langle D_{KL}(p(Y|X)||p(Y|Z)) \rangle_{p(X,Z)}$$
$$= \int p(X,Z)(\int p(Y|X) \log \frac{p(Y|X)}{p(Y|Z)} dY) dX\, dZ$$
$$= \int p(Y,X,Z) \log(\frac{p(Y|X)p(X)}{p(X)p(Y)} \frac{p(Y)p(Z)}{p(Y|Z)p(Z)}) dY dX\, dZ$$
$$= \int p(Y,X,Z) \log(\frac{p(Y,X)}{p(X)p(Y)} dY dX dZ - \int p(Y,X,Z) \log(\frac{p(Y)p(Z)}{p(Y,Z)} dY dX\, dZ$$
$$= I(X,Y) - I(Z,Y)$$

To complete the proof, we need to show that $I(X,Y|Z) = I(X,Y) - I(Z,Y)$. This can be shown by considering the definition of mutual information as

$$I(X,Y) = H(Y) - H(Y|X)$$
$$I(Z,Y) = H(Y) - H(Y|Z) \,.$$

Using the Markov relation $H(Y|X,Z) = H(Y|X)$ we see that $I(X,Y|Z) = H(Y|Z) - H(Y|X,Z) = H(Y|Z) - H(Y|X) = H(Y) - H(Y|X) - H(Y) + H(Y|Z) = I(X,Y) - I(Z,Y)$. $\square$

Since $I(X,Y)$ is independent of $Z$, minimizing the KL divergence happens when $I(Z,Y)$ is maximized, with zero KL being obtained when $I(Z,Y) = I(X,Y)$. In other words, we postulate that a goal of the time series model and the machine learning algorithm is finding a latent representation $Z$ that "explains out" most of the time series information dynamics $I(X,Y)$. This principle is expressed as the minimization of $I(X,Y|Z)$, i.e., finding a latent $Z$ so that there will be very little remaining information passing between the past $X$ and the present $Y$ of the time series observations data themselves.

To complete our model, some additional constraints on $Z$ need to be specified, since if $Z = X$, this minimization condition is trivially satisfied. Accordingly, in the next paragraph, we add constraints that require the latent variable representation to be as compact or as simple as possible. In the following sections we will use variational autoencoding (VAE) as a possible representation learning method. In the process of VAE learning, the latent representation is regularized by bringing its distribution to be as close as possible to a Gaussian uncorrelated noise. Additionally, in this study, we introduce an additional step of lossy compression of $Z$ given a pre-trained VAE. This step will be accomplished using a bit allocation procedure from rate–distortion theory.

### 2.1. Adding Simplicity Requirement of the Latent Representation

As mentioned above, our formulation of the time series modeling goal is to find a meaningful latent $Z$ that best approximates the next observation $Y$. Since minimizing KL could be trivially satisfied by taking $Z = X$, in order to avoid such a trivial solution, we add a constraint on $Z$ by requiring it to be the most compact or simplest latent "explanation" derived from the observation $X$. In information theoretical terms, we can write the criteria as a minimization of $I(X,Z)$. In terms of coding, we look for the least amount of bits of information about $X$ to be contained in $Z$. We also need a third constraint to prevent $I(X,Z)$ from going to zero. This is achieved by adding a fidelity requirement or bound on distortion between $X$ and $Z$, denoted as $D(X,Z)$. It is important to note that this distortion $D$ is not the same as a KL divergence; rather, it is some physically motivated distortion, such as mean square error (MSE) or some other fidelity measure between the "compressed" representation of $X$, as expressed by $Z$, and the original data $X$. For the moment, we will ignore this fidelity constraint and only consider the first competing relations between maximizing $I(Z,Y)$ and minimizing $I(Z,X)$. Combining the two goals, we arrive at the target function for our learning system

$$\max_{P(Z|X)} \{I(Z,Y) - \lambda I(X,Z)\} \tag{6}$$

This formulation bears a close resemblance to the idea of the information bottleneck (IB) [5]. The formulation of the IB is to say that a goal of a learning system is to find the most compact representation $Z$ of $X$ that still provides most information about a different variable $Y$. Accordingly, a predictive IB looks at a combination of factors when predicting the next observation $Y$ from $Z$, which can be solved separately from the encoding of $X$ by $Z$. In the following, we will use the symbolic dynamics of the latent representation to solve the predictive IB criteria using VAE representation learned from $X$.

To summarize what we have discussed so far, we presented the following competing criteria for our time series model, combining three factors $I(X,Z)$, $I(Y,Z)$, and $D(X,Z)$

- Finding a compact representation $Z$ of $X$ from which $X$ can be recovered with minimal distortion $D(X,Z)$ (i.e., reconstruction qualify);

- Finding a representation $Z$ of past data $X$ that is most informative about the next sample $Y$ (i.e., time information).

*2.2. Latent Information Rate*

A central goal of representation learning that stems from the information dynamics principle is maximizing the amount of information passing between the encoded past $Z$ and the next data sample $Y$. Using the second set of Markov relations shown in Figure 4, we would like to consider here the information passing between the latent variables $Z$ and $T$ themselves. We express $I(Z, Y) = I(Z, T) - I(Z, T|Y)$ as a measure of the ability to predict the future of the time series, and the next observation $Y$ from past embedding $Z$, compared with the information dynamics of the latent embedding time series itself. From these information relations, we see that the amount of information that past encoding $Z$ carries out about the future observation $Y$ is less than the amount of information carried between the past and future of the embeddings $Z$ and $T$ themselves. The term $I(Z, T)$ can be considered as latent predictive information, or latent information dynamics, corresponding to some sort of a latent anticipation.

Ignoring $I(Z, T|Y)$ (or assuming it is zero) means that the next data observation $Y$ causes the latent representations of the past and present to be independent. In such a case, the information $I(Z, Y)$ contained in the latent encoding of the past $Z$ about the present data $Y$ is the same as the information contained in the past and present data observations $X$ and $Y$ directly. In such a case, the maximization of the information dynamics in the latent space $I(Z, T)$ is sufficient for maximizing the predictive ability of the model. This assumption allows us to formulate yet another more tractable version of the predictive latent bottleneck problem

$$\max_{P(Z|X), P(T|Y)} \{I(Z, T) - \lambda I(X, Z)\} \tag{7}$$

where the encoding of the past and the present observations are represented by probability distributions $P(Z|X)$ and $P(T|Y)$, respectively. In the case when the past is encoded data point by data point independently in time, the same encoder, and thus same probability function $P(x, z) = P(y, t)$, is used throughout the whole time series. In such a case, we can separately learn an encoder over the whole time series first; then, in the next step, we can evaluate the information dynamics of the sequence of latent variables. When performed in a iterative manner by computing the predictive information using symbolization and further altering an existing encoding using rate–distortion, an optimal final model can be found by an exhaustive search over the bit-rate and quantization threshold parameters.

## 3. Symbolization Dynamics

Given a time series or multi-variable observations of a signal $X = x_1, x_2, \ldots, x_n, \ldots, x_T$, the goal of symbolization is to find a sequence $S = s_1, s_2, \ldots, s_n, \ldots, s_T$ of the same length $T$, where each observation $x_n$ is labeled by a symbol $s_n$ coming from a finite-sized alphabet $s_i \in \Sigma$. The essential step in the symbolization algorithm described below is finding a threshold value $\theta$ that partitions the space of observations into categories based on repeated sub-sequences of motifs. For each new observation, a threshold $\theta$ is used to determine if the incoming $x_n$ is sufficiently similar to an observation that appears as a continuation to an earlier-identified repeated sub-sequence pointed to by a suffix link from a previous step $n - 1$. The recursive procedure of finding repeated suffixes is based on a string-matching algorithm called Factor Oracle (*FO* hereafter), which will be explained next. Using a repeated suffix link, the algorithm assigns two data points that appear as continuations of the two linked sub-sequences, $x_i$ and $x_j$, and the same label $s_i = s_j \in \Sigma$ if $||x_i - x_j|| \leq \theta$. In extreme cases, setting $\theta$ too low leads to assigning different labels to every observation in $X$, while setting $\theta$ too high leads to assigning the same label to every observation in $X$.

The basis for the selection of a threshold is an analysis of the predictive properties of the suffix structure (also called the oracle structure) found at each threshold value. We

define *information rate* (*IR* hereafter) as a measure that is used to select the optimal $\theta$ for an individual time series. We show an example of the oracle structure with extreme $\theta$ values in Figure 5. It should be noted that the alphabet of the symbolization is constructed dynamically, as new symbols are added when an input sample cannot be assigned to one of the existing clusters that were labeled earlier. We will denote the resulting alphabet for a given $\theta$ as $\Sigma_\theta$.

In the process of *IR* analysis, the system is performing a search over different $\theta$ values, where for each threshold value, a different suffix structure is constructed for each resulting symbolization of the same time series, as shown in Figure 5. To select the symbolization with the most informative oracle structures, *IR* is used to measure the average relative reduction of uncertainty of the current sample in a time series when past samples are known.

Let the past samples of a time series be denoted by $x_{\text{past}} = \{x_1, x_2, \ldots, x_{n-1}\}$; the current samples $x_n$ and $H(x) = -\sum P(x) \log_2 P(x)$ with the entropy of $x$ with $P(x)$ and the distribution of $x$; and the statistical definition of *IR* is the mutual information between $x_{\text{past}}$ and $x_n$,

$$I(x_{\text{past}}, x_n) = H(x_n) - H(x_n | x_{\text{past}}). \tag{8}$$

In Ref. [6], the above statistical definition of *IR* was replaced by an algorithmic notion of compression gain using a measure of code length $C(\cdot)$ instead of the entropy term $H(\cdot)$ in (8). The *IR* algorithm searches over all possible values of the quantization threshold $\theta$ (9) to find the highest *IR* value

$$IR(x_{\text{past}}, x_n) = \max_{\theta, s_t \in \Sigma_\theta} [C(s_n) - C(s_n | s_{\text{past}})]. \tag{9}$$

The *IR* measure has been used extensively in the analysis of music information contents, known as music information dynamics [7,8].

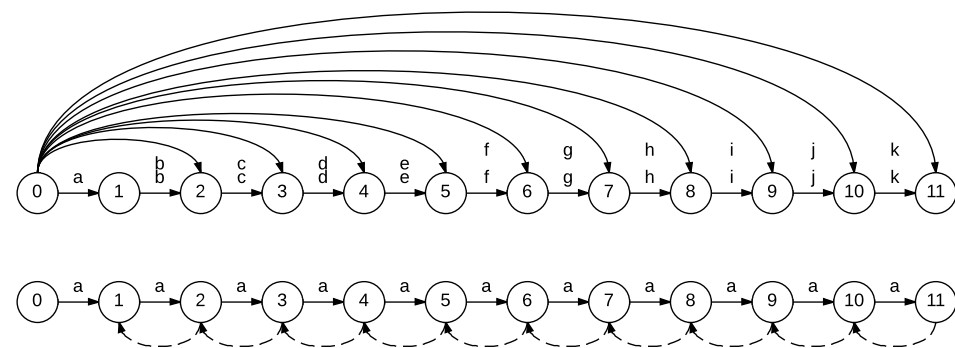

**Figure 5.** Two oracle structures with extreme values of $\theta$. The letters near each forward links represent the assigned labels. (**Top**) The oracle structure with $\theta = 0$ or extremely low. (**Bottom**) The oracle structure with a very high $\theta$ value. It is obvious that in both cases the oracles are not able to capture any structures of the time series.

The value of the algorithmic *IR* defined in (9) can then be robustly calculated by complexity measures associated with a compression algorithm with $C(s_n)$—the number of bits used to compress $s_n$ independently—and $C(s_n | s_{\text{past}})$—the number of bits used to compress $s_n$ using $s_{\text{past}}$. In Ref. [9], a lossless compression algorithm, *Compror*, based on *FO*, is provided. The detailed formulations of how *Compror* and *IR* are combined are provided in Ref. [6]. In the context of time series patterns and structure discovery with *VMO*, the *VMO* with a higher *IR* value indicates that more of the repeating sub-sequences (ex. patterns, motifs, themes, gestures, etc.) are captured than the ones with a lower *IR* value.

### 3.1. Variable Markov Oracle

Here, we present a generalization of Factor Oracle (FO) algorithm to the case of a time series with approximate repeated sequence discovery that we called *Variable Factor Oracle* (*VMO* hereafter). An example of FO structure in the format of VMO for a sequence of labeled states is shown in Figure 6. More examples and details for this structure are demonstrated in Ref. [10].

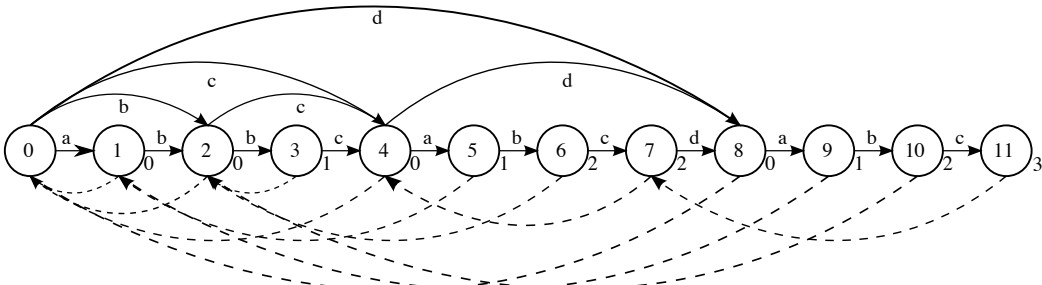

**Figure 6.** A *VMO* structure with symbolized signal $\{a, b, b, c, a, b, c, d, a, b, c\}$. Upper (normal) arrows represent forward links with labels for each frame and lower (dashed) are suffix links. Values outside of each circle are the *lrs* value for each state.

Given a sequence of labels, forward links are used to retrieve any of the sub-sequences from *X*. An oracle structure has two types of forward links. The first is an internal forward link, which is a pointer from state $t - 1$ to $t$ labeled by the symbol $x_t$, denoted as $\delta(t - 1, x_t) = t$. The other forward link is an external forward link, which is a pointer from state $t$ to $t + k$ labeled by $x_{t+k}$ with $k > 1$. An external forward link $\delta(t, x_{t+k}) = t + k$ is created when

$$x_{t+1} \neq x_{t+k}$$
$$x_t = x_{t+k-1}$$
$$\delta(t, x_{t+k}) = \varnothing.$$

In other words, an external forward link is created between $x_t$ and $x_{t+k}$ when the pair of symbols $\{x_{t+k-1}, x_{t+k}\}$ is seen for the first time in *X* and with $x_t$ being linked to $x_{t+k-1}$ via suffix links, thus both sharing the same label.

An oracle structure carries two kinds of links: forward links and suffix links. A suffix link is a backward pointer that links state $t$ to $k$ with $t > k$ without a label and is denoted by $sfx[t] = k$.

$$sfx[t] = k \iff \text{the longest repeated suffix of}$$
$$\{x_1, x_2, \ldots, x_t\} \text{ is recognized in } k.$$

Suffix links are used to find the longest repeated suffix in *X*. In order to track the longest repeated suffix at each time index *t*, the length of the longest repeated suffix at each state *t* is denoted by $lrs[t]$ and is computed by the algorithm described in Ref. [11]. The parameter *lrs* is part of the on-line construction algorithm of the oracle automaton [11].

### 3.2. Symbolization Algorithm

It should be noted the the oracle structure found in the previous section contains $T + 1$ states $Oracle(o_0, o_1, o_2, \ldots, 0_T)$, with the time series observation located along the direct transitions between the states, which are also called internal forward links. Direct links are links to the immediate next state to the right (next step in time) of the oracle structure. For example, $s_1$ is a label of the transition between $o_0$ and $o_1$, and $s_t$ is between $o_{t-1}$ and $o_t$. Once the construction of the oracle structure is completed, the symbolization of the time series is described in Algorithm 1.

---

**Algorithm 1** Symbolization

---

**Require:** $Oracle(o_0, o_1, o_2, \ldots, o_T)$

Following the first direct link and all forward links from state $o_0$, assign a unique symbolic label $s_i \in \Sigma$ to each of the links.

**for** $n = 1 : T$ **do** Follow the suffix link from state $o_n$.
    **if** forward external link exists from $sfx(o_n)$ **then**
        label $s_n$ by the symbol that was used to label that link,
    **else**
        label $s_n$ by the symbol of the direct internal link from $sfx(o_n)$.
    **end if**
**end for**

---

To summarize, the VMO algorithm comprises a combination of the following innovations:

- Extension of the FO algorithm to operate on time series with an approximate matching up to a given threshold;
- Symbolization of the time series according to forward links emerging from state zero. Forward links from state zero point to places in the time series where novel data observations were first encountered and these observations could not be included into continuations of existing suffixes since their distance from previously labeled observations exceeded a set threshold;
- Use of Compror to estimate the information rate at each threshold by considering the difference in coding length without and with Compror;
- Selection of the oracle with the highest information rate and using its symbolization as the best discrete representation of the time series.

By using an exhaustive search over a range of threshold values, an oracle with the highest information rate is retained as the optimal representation.

*3.3. Query-Matching with VMO*

For various purposes to be discussed later in this paper, such as applications of symbolization for signal matching or transfer entropy estimation, there is a need to find a recombination of factors in the original time series that is closest to another time series that is provided as a query into the oracle structure. The query-matching algorithm tracks the progress of traversing the oracle using forward and backward links, finding the optimal path via a dynamic programming algorithm. For full details, the reader is referred to Ref. [12]. We provide the pseudo-code in Algorithm 2 for the sake of completeness.

---

**Algorithm 2** Query-Matching

---

**Require:** Target signal in *VMO*, $Oracle(S = s_1, s_2, \ldots, s_T, X = x_1, x_2, \ldots, x_T)$ and query time series $R = r_1, r_2, \ldots, r_N$

Get the number of clusters, $M \leftarrow |\Sigma|$
Initialize cost vector $C \in \mathbb{R}^M$ and path matrix $P \in \mathbb{R}^{M \times N}$.
**for** $m = 1 : M$ **do**
    $P_{m,1} \leftarrow$ Find the state, $t$, in the $m$th list from $\Sigma$
        with the least distance, $d_{m,1}$, to $r_1$
    $C_m \leftarrow d_{m,1}$
**end for**
**for** $n = 2 : N$ **do**
    **for** $m = 1 : M$ **do**
        $P_{m,n} \leftarrow$ Find the state, $t$, in lists with labels
            corresponding to forward links from state
            $P_{m,n-1}$ with the least distance, $d_{m,n}$ to $R[n]$
        $C_m \mathrel{+}= d_{m,n}$
    **end for**
**end for**
**return** $P[(C)], \min(C)$

---

### 4. Representation

As mentioned earlier in the chapter, a suitable choice of time series representation is critical for the analysis of time series. Transforms, such as DFTs and wavelets, or adaptive transforms, such as SVD, are commonly used to find an embedding of the signal in another space that better captures some salient aspect of the data. In other cases, feature extractors are designed to represent the time series in terms of the special parameters of interest. The symbolization technique presented above is feature agnostic and can be applied to any sequence of features if an appropriate distance function is provided to measure feature or embedding similarity. To avoid the step of feature engineering or the choice of appropriate transforms, it would be desirable to have a method that allows learning a representation with sufficient quality and predictability according to the information dynamics objective that was defined in Section 2.

Recently, generative deep neural models have become the primary tools for representation learning in multiple domains. The compelling idea of representation learning is that various aspects of modeling data from the world are captured using vector spaces that map statistical structures into mathematical space. The information dynamics objective states the goal of representation learning as a minimization of the problem of the mutual information between the data and a latent state, namely $min_{P(Z|X)} I(X, Z)$, which can be subject to a distortion constraint $D(X, Z)$. Such an approach is also known as a rate–distortion problem in information theory.

The relation between variational encoding and rate distortion was made explicit in the "Broken ELBO" paper [13]. The authors show there that the mutual information between the input $X$ and the latent code $Z$ is bounded below and above by two factors $D$ and $R$ that comprise the ELBO, as follows

$$H - D \leq I_e(X, Z) \leq R \qquad (10)$$

where $H$ is the data entropy, $D$ is the distortion measured as reconstruction log likelihood, and $R$ is the model encoding rate (not the optimal Shannon rate, which is the mutual entropy) measured by KL divergence between the encoding distribution and the learned marginal approximation. The reader is referred to the appendix for a short summary of the results, and to Ref. [13] for more complete detail. The important point here is that this expression can be rewritten as a Lagrange optimization by adding $D$ to all terms or the above inequality, giving

$$H \leq I_e(X, Z) + D \leq R + D = -ELBO. \qquad (11)$$

Moreover, allowing for different weight of the distortion $D$, we have

$$I_e(X, Z) + \beta D \leq R + \beta D = -ELBO(\beta). \qquad (12)$$

where $\beta = 1$ gives the original VAE ELBO expression and $ELBO(\beta)$ corresponds to $\beta$-VAE model. This expression is an unconstrained version of the rate distortion,

$$\mathcal{L} = I(X, Z) + \beta \langle d(X, Z) \rangle \qquad (13)$$

where minimizing for $\beta$ value gives a point on the rate–distortion curve.

Changing $\beta$ is important for dealing with two known problems in VAE encoding: Information Preference and Exploding Latent Space problems. The first refers to vanishing of the mutual information between $Z$ and $X$, or in other words $Z$ and $X$ becoming independent due to a powerful decoder. The second problem refers to over-fitting of the output likelihood by matching individually each data point when the training data is finite. #This also effectively causes the latent states to become irrelevant.

One of the approaches proposed in this paper is applying the variational autoencoder (VAE) [14] as an embedding of the time series into a sequence of latent variables. Unlike

standard autoencoders that translate data into a more desirable latent representation found in the activation of the hidden layers, VAE explicitly constrains the latent variables $Z$ so that they should be random variables distributed according to some prior $p(z)$. The input $X$ and latent code $Z$ can then be seen as jointly distributed random variables $Z \sim p(Z), X \sim p(X|Z)$. The VAE consists of an encoder probability $q_\lambda(Z|X)$, which approximates the posterior probability $p(Z|X)$, and a decoder probability $p_\theta(X|Z)$, which parametrizes the likelihood $p(X|Z)$. In practice, the approximate posterior and likelihood distributions are parametrized by the weights of connections between neurons in the VAE network. Posterior inference is performed by minimizing the KL divergence between the encoder and the true posterior. It can be proved that this optimization problem is same as maximizing the evidence lower bound (ELBO):

$$ELBO = E[\log p_\theta(X|Z)] - KL(q_\lambda(Z|X)||p(Z)) \leq \log p(X) \tag{14}$$

Another important insight about VAE is that it is a generative model, where the latent variables $Z$ are used to "drive" the decoder into producing novel samples of data $X$. In a statistical sense, VAE is used to learn the distribution $P(X)$ rather then simply encode the observations $X$.

### 4.1. Rate Reduction

Rate–Distortion is studied as a way of extracting useful or meaningful information from noisy signals, and is also motivated by aspects of human cognition [15]. Reduced representation was also explored in the context of deep neural network learning theory using the information bottleneck principle [5]. In deep learning, some attempts to consider predictive information through the use of a bottleneck or noisy representation in temporal models, such as RNNs have recently appeared in the literature[16,17]. An important distinction between these works and the proposed framework is that these works are using rate limitation as part of the learning process. Here, we propose applying bit reduction to a pre-trained encoder–decoder network in order to reduce the complexity of the latent representation prior to decoding. This allows experimenting with various bit-rate regimes for a fixed embedding without retraining the representation network. For this purpose, we borrow bit allocation technique from the rate–distortion theory of lossy information processing. The reduced latent representation is subjected to information rate analysis using symbolization methods as previously explained.

### 4.2. Bit-Rate-Limited Encoding

As mentioned above, in this study we consider a particular case of reduced representation that is based on rate–distortion theory. Rate–distortion theory offers an optimal solution for finding the most compact (least-rate) encoding for a given limit on the distortion or reconstruction error. Equivalently, distortion–rate finds the best encoding in terms of least distortion for a given rate. Algorithms for optimal bit allocation according to the rate–distortion theory are so called bit allocation methods that we describe below. By using a rate-limited channel between the encoder and decoder of the VAE, we are able to control the complexity of the encodings using a bit allocation algorithm. In our case, we use rate as the free parameter to find the least-distortion codes under the assumption that latent codes in VAE are distributed as multi-variate uncorrelated Gaussians. The rate–distortion function that provides the lower limit on the achievable rate $R$ as a function of the maximal-allowed distortion level $D$, is given by

$$R(D) = \begin{cases} \frac{1}{2}log_2\frac{\sigma^2}{D}, & \text{if } 0 \leq D \leq \sigma^2 \\ 0, & \text{if } D > \sigma^2. \end{cases} \tag{15}$$

where $R$ is the rate and $D$ is the distortion value. This rate–distortion function can be converted into a distortion–rate function $D(R) = \sigma^2 2^{-2R}$ that gives the lower limit on distortion D that is achievable for a given rate R. This ideal lower limit (i.e., least distortion)

can be efficiently achieved for a particular type of signal that is known in communication theory as a "multivariate Gaussian channel". We adopt this channel model for our experiments without further justification. What is special about this type of channel is that an optimal bit reduction can be achieved by using the so-called reverse water-filling algorithm [18]. This algorithm starts with a predefined bit-rate R and successively allocates one bit at a time to the strongest component, repeating the process until all bits in the bit pool are exhausted. One should note that channels (i.e., latent variables in our case) with variance less than allowed distortion, or channels that run out of bits for a given rate, are given zero bits and, thus, are eliminated from the transmission.

A schematic representation of channel inclusion in the autoencoder architecture is given by Figure 7.

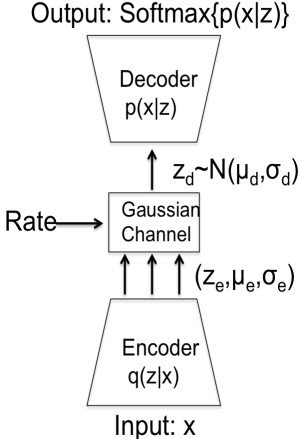

**Figure 7.** Noisy channel between encoder and decoder.

Encoding the latent components at a rate $R$ changes the mean and variance of the VAE as follows [19]

$$Q(z_d|z_e) = Normal(\mu_d, \sigma_d^2) \tag{16}$$

$$\mu_d = z_e + 2^{-2R}(\mu_e - z_e) \tag{17}$$

$$\sigma_d^2 = 2^{-4R}(2^{2R} - 1)\sigma_e^2 \tag{18}$$

This process requires some explanation: for a given rate $R$ we obtain the bit-rate for each of the latent variables according to the reverse water-filling procedure. This gives us different numbers of bits for each latent dimension, where the high-variance dimensions tend to grab the bits first, often leaving the weak (small-variance) latent variables with zero bits. Next, we proceed by sampling a value from the encoder distribution according to the original VAE mean and variance parameters, $\mu_e$ and $\sigma_e^2$. Then, using the rate $R$ and the original mean and variance parameters for each latent variable, we derive a new mean and variance $\mu_d$ and $\sigma_d^2$. We use these probability parameters to sample a reduced-bit-rate value and use it as our new input to the decoder. One can see from Equation (18) that latent variables that are allocated zero bits need not be transmitted. More precisely, the value that the decoder needs is the mean value of that latent variable that is independent of the particular instance being transmitted. This mean value can be obtained a priori and thus can be "hard coded" into the decoder ahead of time, with no need to transmit it. Channels allocated a very high rate will transmit an (almost) unaltered value of the latent variable that was sampled in the VAE encoder.

*4.3. Combined Optimization*

The algorithm for combined predictive and representation optimization that accomplishes the information dynamics objective (7) is given in Algorithm 3. This algorithm

consists of an inner symbolization loop that finds the best quantization threshold for providing latent representation quality, and an outer loop that uses bit allocation to reduce the fidelity of a previously learned encoding. Although not a fully end-to-end representation and prediction method, we propose this algorithm as a way to search over lossy variants of previously learned high-rate representation encoding, taking into account the predictive properties of the encoded time series using symbolic dynamics.

---

**Algorithm 3** Combined Rate-Prediction

---

**Require:** Given a time series $X = (x_1, x_2, \ldots, x_T)$, an element-wise encoder–decoder pair $z = Enc(x), \hat{x} = Dec(z)$, with $Z = (z_1, z_2, \ldots, z_T)$, an error function $Dist(x, \hat{x})$, a bit-allocation procedure with total rate parameter R, $\hat{z} = Bitalloc(z, R)$, and a symbolization oracle with threshold $\theta$, $S = Oracle(Z, \theta)$
Perform an encoding $Z = Enc(X) = (z_1 = Enc(x_1), z_2 = Enc(x_2), \ldots, z_T = Enc(x_T))$
**for** rate $R$ from $R_{max}$ to $R_{min}$ with decrements $\delta$ **do**
    Compute the reduced rate latent code $\hat{Z} = Bitalloc(Z, R)$
    Compute the mean error $D = E(Dist(X, Dec(\hat{Z})))$
    Normalize the sequence Z so that the maximal distance between samples of Z is 1
    **for** threshold $\theta$ from 0 to 1 with increments $\epsilon$ **do**
        Compute $\theta^* = argmax(IR(\theta))$
        Store the result in a Loss Matrix $L(R, \theta^*) = IR(\theta^*) - \lambda D$
    **end for**
**end for**
For best model select $R^*, \theta^{**} = argmin_{R, \theta*} L(R, \theta^*)$. **return** a symbolic encoding $S$ of the original time series $X$, $S = Oracle(Bitalloc(Enc(X), R^*), \theta^{**})$

---

## 5. Transfer Entropy Estimation

Transfer entropy between two sequences is the amount of information passing from the past of one sequence to another when the dependencies of the past of the other sequence (sequence's own dynamics) have been already taken into account. To distinguish the notation from that of previous sections, we will denote the two time series as data $X$ and context $C$, and their past as $\bar{X}$ and $\bar{C}$, respectively. This gives $TE_{C \to X} = I(X; \bar{C}|\bar{X})$ Similarly $TE_{X \to C} = I(C; \bar{X}|\bar{C})$. Writing mutual information in terms of entropy

$$I(C; \bar{X}) = H(C) - H(C|\bar{X})$$
$$I(C; \bar{X}|\bar{C}) = H(C|\bar{C}) - H(C|\bar{X}, \bar{C})$$

Adding and subtracting $H(C)$:

$$I(C; \bar{X}|\bar{C}) = H(C|\bar{C}) - H(C|\bar{X}, \bar{C}) - H(C) + H(C) = I(C; \bar{X}, \bar{C}) - I(C; \bar{C}) \quad (19)$$

Additionally:

$$I(X; \bar{C}|\bar{X}) = I(X; \bar{C}, \bar{X}) - I(X; \bar{X}) \quad (20)$$

We consider a sum of these expressions, calling it symmetrical transfer entropy (SymTE):

$$SymTE = I(C; \bar{X}|\bar{C}) + I(X; \bar{C}|\bar{X}) \quad (21)$$

**Lemma 2.** *Given time series X and C, the symmetrical transfer entropy is given by*

$$SymTE = I((C, X); \overline{(C, X)}) - I(C; X|\overline{(C, X)}) + I(C, X) - I(X, \bar{X}) - I(C, \bar{C}), \quad (22)$$

*where we used a notation for the past of the joint pair $(\bar{C}, \bar{X}) = \overline{(C, X)}$.*

**Proof.** Symmetric transfer entropy (SymTE) between two sequences $X$ and $C$,

$$SymTE = I(C; \bar{X}|\bar{C}) + I(X_i; \bar{C}|\bar{X})$$

equals to

$$SymTE = I((C, X); \overline{(C, X)}) - I(C; X | \overline{(C, X)}) + I(C, X) - I(X_i, \bar{X}) - I(C, \bar{C}) ,$$

where we used a notation for the past of the joint pair $(\bar{C}, \bar{X}_i) = \overline{(C, X_i)}$. Using the relation

$$I(X; \bar{C} | \bar{X}) = H(X | \bar{X}) - H(X | \bar{X}\bar{C}) = H(X | \bar{X}) - H(X) + H(X) - H(X | \bar{X}\bar{C}) = I(X; \bar{C}\bar{X}) - I(X, \bar{X})$$

and similarily

$$I(C; \bar{X} | \bar{C}) = I(C; \bar{X}\bar{C}) - I(C, \bar{C})$$

We sum both of the above expressions to derive the symmetrical TE:

$$SymTE = I(C; \bar{X} | \bar{C}) + I(X; \bar{C} | \bar{X}) = I(C; \bar{X}\bar{C}) - I(C; \bar{C}) + I(X; \bar{X}\bar{C}) - I(X; \bar{X})$$
$$= I(C; \bar{X}\bar{C}) + I(X; \bar{X}\bar{C}) - I(C; \bar{C}) - I(X; \bar{X})$$

Continuing the derivation

$$I(C; X) = H(C) + H(X) - H(C, X)$$
$$I(C; \bar{X}\bar{C}) = H(C) - H(C | \bar{X}\bar{C})$$
$$I(X; \bar{X}\bar{C}) = H(X) - H(X | \bar{X}\bar{C})$$
$$I(CX; \bar{X}\bar{C}) = H(C, X) - H(C, X | \bar{X}\bar{C}) = -I(C, X) + H(C) + H(X) - H(C, X | \bar{X}\bar{C})$$
$$= -I(C, X) + H(C) + H(X) - H(C, X | \bar{X}\bar{C}) - H(C | \bar{X}\bar{C}) + H(C | \bar{X}\bar{C}) - H(X | \bar{X}\bar{C}) + H(X | \bar{X}\bar{C})$$
$$= -I(C, X) + H(C) - H(C | \bar{X}\bar{C}) + H(X) - H(X | \bar{X}\bar{C}) - H(C, X | \bar{X}\bar{C}) + H(C | \bar{X}\bar{C}) + H(X | \bar{X}\bar{C})$$
$$= -I(C, X) + I(C, \bar{X}\bar{C}) + I(X, \bar{X}\bar{C}) + I(C, X | \bar{X}\bar{C})$$

This gives the equality:

$$I(C, \bar{X}\bar{C}) + I(X, \bar{X}\bar{C}) = I(CX, \bar{X}\bar{C}) - I(C, X | \bar{X}\bar{C}) + I(C, X)$$

Plugging back into SymTE gives:

$$SymTE = I(C, \bar{X} | \bar{C}) + I(X, \bar{C} | \bar{X})$$
$$= I(C, \bar{X}\bar{C}) + I(X, \bar{X}\bar{C}) - I(C, \bar{C}) - I(X, \bar{X})$$
$$= I(CX, \bar{X}\bar{C}) - I(C, X | \bar{X}\bar{C}) + I(C, X) - I(C, \bar{C}) - I(X, \bar{X})$$

□

If we assume that the generation of $X$ is independent of $C$ given their joint past $\overline{(C, X)}$, then $I(C; X | \overline{(C, X)}) = 0$, resulting in

$$SymTE \approx I((C, X); \overline{(C, X)}) - I(X, \bar{X}) - I(C, \bar{C}) + I(C, X) , \tag{23}$$

which is a sum of the *IR* of the joint pair $(C, X)$ and the mutual information between $C$ and $X$ regardless of time, minus the *IR* of the separate time series. In other words, the symmetrical TE is a measure of surprisal present in the joint time series minus the surprisal of each of its components, plus the mutual information (lack of independence) between their individual components. This captures the difference between surprisal when considering the compound time series versus surprisal when considering them separately, with an added component of mutual information between the observations of the two time series regardless of time.

*5.1. Border Cases*

If $C = X$, and since $H(X, X) = H(X)$, we obtain $I((X, X); \overline{(X, X)}) = IR(X)$ and $SymTe = I(X, X) - IR(X) = H(X) - H(X) + H(X | \bar{X}) = H(X | \bar{X})$, which is the conditional entropy of $X$ given its past. So, the TE of a pair of identical streams is its entropy rate. If $C$

and $X$ are independent, then $SymTE = 0$. This is based on the ideal case of the *IR* estimator of the joint sequence $I((C, X); \overline{(C, X)})$ being able to reveal the *IR* of the individual sequences and additionally capture any new emerging structure resulting from their joint occurrence. In theory, if $C$ and $X$ are independent, $H(C, X) = H(C) + H(X)$ and $H(C, X | \overline{(C, X)}) = H(C|\bar{C}) + H(X|\bar{X})$; therefore, $I((C, X); \overline{(C, X)}) = I(X, \bar{X}) + I(C, \bar{C})$. Thus, a combination of two time series may add additional information; however, in practice, it could be that VMO will not be able to find sufficient motifs or additional temporal structures when such a mix is performed. In such a case, it can be that the $SymTE$ estimate becomes negative.

*5.2. Mutual Information Neural Estimation*

Theoretically, computing the mutual information between two variables is a hard problem. The computation is only tractable if two variables are discrete or if two variables' probability distributions are known. For the latent variables sampled from a VAE, we cannot directly compute the mutual information since it is intractable to marginalize the probabilities from the whole latent space.

Mutual information neural estimation (MINE) [20] is a framework that allows us to use a neural network output to approximate the mutual information between two variables. Suppose that we want to estimate the mutual information between $X$ and $C$, we first construct the joint data points $(x_1, c_1), (x_2, c_2), ...., (x_N, c_N)$, where for each $n$, the tuple $(x_n, c_n)$ hypothesizes that these two variables are related. Then, we create a new group of data points $(x_1, c_{a_1}), (x_2, c_{a_2}), ...., (x_N, c_{a_N})$, where $(a_1, a_2, ..., a_N)$ is a randomly shuffled index sequence from 1 to $N$. Next, we feed these two groups of data points into a neural network $T_\theta$ to converge by a loss function:

$$L(\theta) = \frac{1}{N} \sum_{n=1}^{N} T_\theta(x_n, c_n) - log(\frac{1}{N} \sum_{n=1}^{N} e^{T_\theta(x_n, c_{a_n})}) \tag{24}$$

where minimizing this loss is able to find a tighter lower bound that can approximate the true mutual information using the Donsker–Varadhan representation of the KL divergence. Generally, the mutual information estimation is performed by finding a mapping between two sets of data points, so it is a "point-wise" mutual information that is not easily applicable to time series or predictive information. In the following experiments, we will use MINE to analyze the mutual information between the latent variables of the generative model data $X$ versus the context $C$.

## 6. Applications

In this section, we mention some of the applications of the information dynamics analysis and the use of symbolization and rate-reduction methods in the areas of audio and other time series analysis. These works point to the significant potential of using symbolic dynamics to capture non-linear dynamical statistics in complex time series and sequences of signal features.

*6.1. Information Rate and Volatility*

The intuition behind information dynamics can be seen by examining how *IR* corresponds to volatility in a financial time series. In Figure 8, the volatility in stock prices is shown together with an instantaneous information rate plot. The data are taken from the daily returns of the S&P500 stock index from 1 October 1983 to 30 August 1991. The volatility of the series was estimated by taking the root of the five-day average of squared returns, as described in Ref. [21]. One should note that the application of information rate analysis is performed here on a short time basis, depending on the length of the repeated suffix at each data point in time.

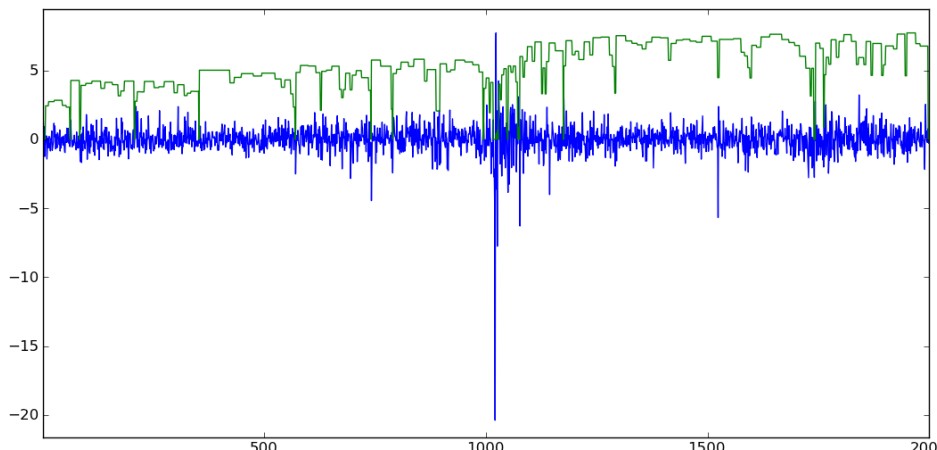

**Figure 8.** S&P500 stock prices shown together with information rate graph.

One can visually observe that regions of extreme change in the stock price correspond to drops in the instantaneous information rate graph. When changes in the stock index are unpredictable, or in other words, when patterns of price change cannot be attributed to the repetition of previous sub-sequences, the information rate significantly diminishes. Such a measure can be used for change detection or segmentation.

*6.2. Motif Finding*

Discovering musical patterns (motifs, themes, sections, etc.) is an important task for musical analysis, song cover identification, composition, and more. One can also consider this as a general time-series problem of identification of salient repeated sub-sequences that might be important for the characterization of data. In music the task of motif finding is defined as identifying salient musical patterns that approximately repeat within a piece. These patterns could potentially be overlapping with each other, and their repetitions could be inexact in terms of variations of various musical parameters such as harmony, rhythm, melodic contours, etc.

In Ref. [7], the VMO algorithm was used to discover patterns in audio recording using a special feature called a *chromagram* that aggregates the signal energy from different frequencies into 12 bins that correspond to 12 basic musical pitches (pitch classes). The chromagrams were specially processed so as to match the beat structure of the music. To consider transposition, the distance function used in the VMO is replaced by a cost function with transposition invariance. The algorithm for pattern discovery is presented in Ref. [7]. The only parameter it requires is a minimum pattern length. The VMO performance was SOTA compared with other existing methods as tested on a musical benchmark that was produced by human experts as the ground truth for motif analysis. Further uses of this method allowed us to construct motif-grams (plots of repeated motifs that occur in music over the time course of the musical piece). An example of a motif-gram for a recording of a Bach musical piece is shown in Figure 9.

The x-axis corresponds to time, counted in terms of frames of audio analysis, and the y-axis shows the motif count with black horizontal lines corresponding to the time frames during which the motif appeared. In this specific case, the special chroma feature that is designed to detect tonal or chord-like patterns in music, invariant to the exact notes that are being played, found the most repeating motifs. MFCC features that capture the sound timbre or color show the least amount of repetitions, indicating that changes in sound color are not a significant structural feature in this musical piece. The VAE representation finds an intermediate amount of motifs; however, it is not the optimal feature for this type of data. This example demonstrates the limited ability of representation learning to find very specific or highly sophisticated structures in specialized types of data. Nevertheless, when no human engineering features or prior knowledge is available to find a custom

representation, VAE can still be used as a way to learn a useful representation. These motif-finding methods were applied for a comparative analysis of musical cultures in Ref. [22], showing that the VMO algorithm has "tuned" itself to different levels of quantization in different cultures, thus suggesting that the basic salient elements of musical expression differ according to different cultures. To give a concrete example, comparing the music of Bach or Telemann played on a recorder (wooden flute) to Japanese Shakuhachi music suggested that the latter musical style had many more musical inflections and shorter patterns in sound, such as noise and pitch variations, compared with the two Western composers, who organized their music into longer and more clearly defined note sequences. The query-matching algorithm that was presented above in the context of transfer-entropy estimation has been previously applied for sequence identification in the case of human gestures. In Ref. [12], multiple oracle structures were constructed for human gestures using a manually engineered representation of skeletal positions recorded from a depth camera. Using a count of the number of recombinations of a new query, the oracle with the least number of recombinations was chosen as the closest reference gesture to the new input. This method of gesture classification showed better performance compared with the DTW and HMM models and was on par with SVM trained on special features (covariances between joints as features).

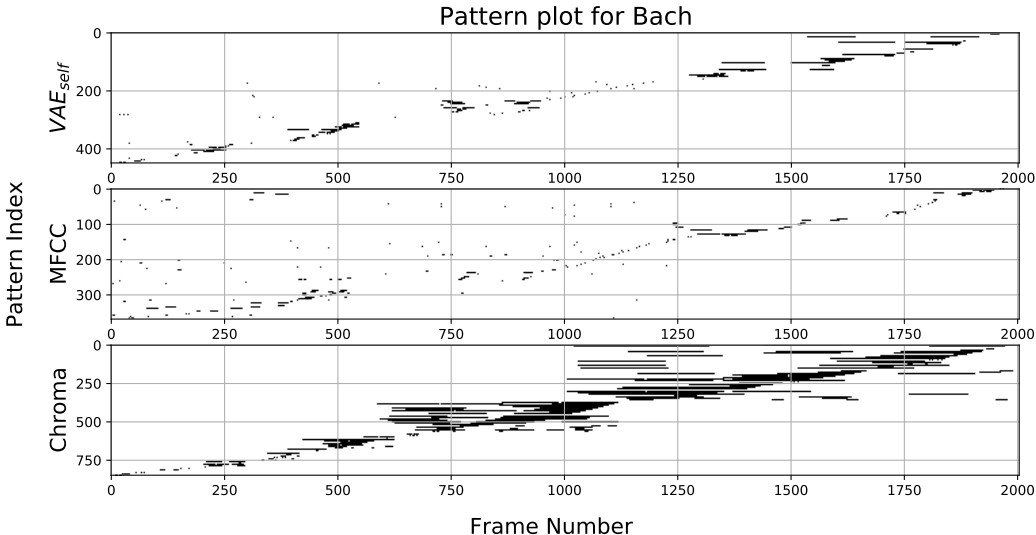

**Figure 9.** Motifs found in Bach recording for different features.

*6.3. Recurrent Quantification Analysis*

Another field of application for symbolic dynamics is non-linear time series analysis (NLTSA). When applied to human voices, NLTA characterizes disorders in voices that are depicted in their temporal dynamics rather then in their instantaneous acoustic properties. Numerous studies try to characterize non-linear phenomena in vocal signals with evidence of non-linear dynamical behavior in speech signals and pathological voices. In a series of works [23,24], we have applied VMO as a pre-processing step for the extraction of non-linear features using so-called recurrent quantification analysis (RQA). RQA extracts features or statistics from a recurrence plot (RP) that is a two-dimensional binary representation of the distances between signal trajectories in phase space over time, where a Heaviside function (0 or 1 mapping) is applied to the distances between points in the phase space at any given pair of time points to determine if it is below or above that distance threshold. Instead of the classical time-delay embedding of the signal, VMO is used for a short-term spectral analysis of the acoustic signal, resulting in optimal symbolic representation that renders itself to a simple recurrent analyzer from its oracle suffix structure. Simply described, a repeated suffix found by the oracle is equivalent to a recurrence diagonal sequence in a RP.The use of symbolization not only automates the process of finding approximate repetition but it

also solves the issue of finding the optimal threshold for generating the RP. The combined VMO-RQA method was applied to various acoustic signals, from the detection of emotions and affect in speech and audio to the detection of COVID-19 in vocalizations of coughs and sustained vowels. Slightly different audio features were extracted in each case (constant Q or CQT transform, and mel frequency cepstral MFCC, in the affect and COVID-19 applications, respectively).

### 6.4. Creative Application

SymTE was used to select between generative models of musical style trained as multiple VMO models. It is beyond the scope of this paper to describe how VMO is used in a generative manner; however, broadly speaking, novel musical data can be generated by the recombination of phrases from an existing musical piece by following the suffix links in an oracle structure found in a VMO analysis of that piece. Such recombinations can be performed in a random manner to create a random improvisation in a given piece of music or to create an accompaniment to another musical piece by using it as a query on a pre-trained oracle. In Ref. [25], the SymTE metric was shown to successfully select the best improvisation model to match a reference musical input, outperforming other sequence-matching methods.

In a different work, rate reduction was used in a creative way to compose a musical piece by encoding–decoding a set of musical pieces using a VAE trained on a different musical piece of a related genre. Details of that work are described in Ref. [26]. A video of the performance can be seen at https://www.youtube.com/watch?v=NSw1XfKuraw. (accessed on 26 November 2022).

### 7. Summary and Discussion

In this paper, we surveyed a set of methods and algorithms that allow an analysis of information dynamics in a continuous time series or in signal data by a process of symbolization. The problem is discussed by addressing the steps of representation learning, embedding by reduction of the representation complexity using lossy coding, and finally the quantization of the continuous embedding into a discrete representation by maximizing the predictive information of the resulting symbolic sequence. The process of quantization is performed by setting a threshold for similarity between the embedding vectors for searching repeated sequences by following a chain of approximately similar suffixes. The resulting suffix–links structure is then used to derive a measure of predictive information by using the suffixes as a way to preform data compression by recopying blocks of current data points from their earlier occurrences. The extent of such compression is then used as a measure of the conditional entropy of the data based on their past. By performing an exhaustive search over possible threshold values, a sequence with maximal difference between its unconditional and conditional entropy, which comprises its predictive information measure, is chosen as the optimal symbolization sequence. The task of designing features for embedding time series data is one of the more challenging steps in this analysis. To address this challenge, applying methods of learning representation by neural embeddings, followed by a step of compression by rate reduction to optimize the temporal structure, can be a significant step towards more fully automating such complex signal analysis tasks. Investigating the trade-off between the fidelity and quality of data representation is formulated in terms of a predictive bottleneck that combines both the symbolic information rate of the latent representation at different encoding rates and the accuracy of the encoding–decoding scheme at each such rate. The survey concludes by showing several applications of symbolization, including an extension to transfer entropy estimation between two parallel sequences. It should be noted that the proposed symbolization method is not assured to achieve a generating partition, for which the topological entropy of a dynamical system achieves its supremum. Nevertheless, we hope that by carefully representing the data and maximizing the predictive qualities of the symbolized approximation, meaningful aspects of the data structure can be efficiently found.

**Funding:** This research received no external funding.

**Data Availability Statement:** Not applicable.

**Conflicts of Interest:** The authors declare no conflict of interest.

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
