# Peer review of "Predictive Quantization and Symbolic Dynamics"

_algorithms, doi:10.3390/a15120484_

Round 1
Reviewer 1 Report
There is an important detail that I think the authors miss. To make a symbolic dynamical analysis requires a good partition. Meaning a generating partition. Without a generating partition, a symbolic dynamics is still induced but it does not completely represent the dynamics, but rather projectively. Therefore drawings such as Fig 1, a communities diagram are not 1-1. This can be a critical issue. See for example,
Bollt, Erik M., et al. "What symbolic dynamics do we get with a misplaced partition?: On the validity of threshold crossings analysis of chaotic time-series." Physica D: Nonlinear Phenomena 154.3-4 (2001): 259-286.
Bollt, Erik M., et al. "Validity of threshold-crossing analysis of symbolic dynamics from chaotic time series." Physical Review Letters 85.16 (2000): 3524.
but I feel the paper is still ok if this issue is discussed and what is and is not observed. This may not require a major rework but rather perhaps two or three paragraphs.
Author Response
Thank you for the review and the important comment about the limited validity of the threshold based partition for analysis of complex and chaotic time series. A discussion was added in the revised version of the paper that clarifies that the proposed symbolization methods is not assured to be a generating partition and that the resulting dynamics consists of a projective representation. References to the papers by Bollt et al. with mention of the problem of threshold-crossing analysis were added in the introduction section, as well as a second mention in a new summary and discussion section that was added at the end of the paper. A comment was added to Fig. 1 pointing to the limited validity of this representation, per reviewer's comment.
Reviewer 2 Report
I find the article interesting and meaningful. In general I do not have much to add other than some of the parts require more explanation, and consistency in terms. The following comments I would like the author to consider in their revised version.
- For a full research paper, too much tangential information is provided (why mentioning GMM?) and I had a hard time to follow the author. As improvement to the paper, which is more like tutorial style or chapter, I would suggest to provide some conceptual diagrams to visualise the often wordy explanations.
- The mathematics are useful, but for the application I would expect some graphs and real examples (which would be expected for a tutorial style paper).
- The mention of motifs is interesting and I would think that using motifs for networks could be a great application, which could be illustrated. The use of recurrence plot quantification analysis is of course suitable, but I am missing a the connection between order pattern (plots) and symbolic dynamics.
- I assume the author means <D_KL> is the distance and D_KL is the KL divergence? There seems to be a bit of a mix up.
- on page 2: the "The structure of the paper is as follows" - is not required and if it was required, the logic does not flow efficiently enough (superfluous information).
- Terms, which are later on used centrally, need to be properly defined; eg "Information dynamics" is mentioned in context with "bottleneck" but not really defined and explained in the introduction. This is just one example, but due to the inconsistent use of words and synonyms, it is sometimes hard to follow (reading becomes unpleasant).
Author Response
I would like to thank the reviewer for his insightful and detailed comments. In response, the following edits were done to the paper.
The paragraph in the introductory section on symbolic discretization was rewritten. Mention of GMM was removed, and a mention of two existing systems, SAX for time series exploration, and VQ-VAE (Jukebox system) were added with diagrams, to demonstrated the concepts and provide a better tutorial coverage of the topic.
To better demonstrate the applications of information rate analysis, an example of analysis of S&P500 stocks time series was added with a graph showing a visual correspondence between volatility and changes (drops) in information rate.
For motif analysis, an example of motif structure in an audio recording of a Bach music piece was provided, comparing different audio features in a figure with explanation.
The notation of <D_KL> was corrected.
The paragraph describing the structure of the paper was rewritten in a more precise and fluid way, I hope.
Terms were checked throughout the paper and better defined as needed (information dynamics, bottleneck, etc).
Round 2
Reviewer 2 Report
I am happy with the authors’ amendments.